# Environmental Compliance and Enterprise Innovation: Empirical Evidence from Chinese Manufacturing Enterprises

**DOI:** 10.3390/ijerph18041924

**Published:** 2021-02-17

**Authors:** Meng Liu, Yun Liu, Yongliang Zhao

**Affiliations:** College of Economics, Jinan University, Guangzhou 510632, China; maxlou1991@163.com

**Keywords:** environmental compliance, enterprise innovation, Chinese manufacturing enterprises, U-shaped relationship

## Abstract

This paper embeds environmental compliance factor and compliance cost factor into the M-O monopolistic competition and multi-product firm model to construct a theoretical model applicable to environmental compliance and enterprise innovation. In addition, we also construct a new environmental compliance index. We use the random-effects Tobit model and the double hurdle model to empirically test the micro-data from the Database of China Industrial Enterprises from 1998 to 2013, then we use the Generalized Propensity Score Matching (GPSM) to conduct a robustness test. The robustness conclusion is that environmental compliance has a significant U-shaped relationship with enterprise innovation, which means, environmental compliance will inhibit enterprise innovation on the left of the inflection point of environmental compliance (0.669), while environmental compliance on the right of the inflection point will promote enterprise innovation. The sub-sample regressions show that, enhanced environmental compliance of state-owned enterprises, mature enterprises, core area enterprises and export enterprises with low level of the environmental compliance, makes the greater inhibition to enterprise innovation, and enhanced environmental compliance of above enterprises with high level of the environmental compliance, makes the greater contribution to enterprise innovation. To this end, the government should adopt the policy, from the shallower to the deeper, to promote the construction of environmental compliance, and identify the inflection point of the environmental compliance on enterprise innovation to stimulate the role of environmental compliance in promoting enterprise innovation.

## 1. Introduction

The sixth “Global Environment Outlook” (GEO-6) put forward by the United Nations in March 2019 claimed that environmental pollution has become a main issue restricting the steady rise of economic and social development. It is difficult to solve the problem of environmental pollution effectively by means of market regulation, the reason is that the environment has the characteristics of public goods. Although environmental regulation can make up for part of the market failure and increase the profitability of enterprises [1], it would inhibit enterprise innovation [2]. Some scholars (e.g., [3]) believe that technological innovation has become the high priority for the market sector to balance environmental protection and economic benefits. Since the reform and opening up, China has experienced a transition period from rapid economic growth to high-quality economic growth and environmental pollution has more and more inhibitory effects in economic development in China. Therefore, it is an urgent task for China to enter into building an innovative country in an all-round way to evaluate comprehensively and reflect the effectiveness of environmental compliance and implement effective adjustment and reform. Hence, whether environmental compliance promotes enterprise innovation or not is the topic of this paper, by aiming at solving the problem that it is difficult for enterprises to balance environmental protection and economic benefits.

According to much scholars, with the strength of environmental regulation increasing, the enterprise innovation tends to rise, decline, decline first and then rise, or insignificant. The first category is the promoting effect. The Porter hypothesis claimed that environmental protection policies will promote enterprises’ technological innovation and rise enterprises’ competitiveness. The positive effects of environmental regulation on the macroeconomy are mainly manifested through the innovative behavior of enterprises [4,5]. Then, some scholars found that the improvement of environmental regulation intensity could significantly promote technological innovation (e.g., [6,7,8,9,10]). The second category is the inhibitory effect. Neoclassical economics believes that environmental protection policies will lead to an increase in the production cost of enterprises, and the positive effects of environmental protection policies on the society will all be offset by the increased production cost of enterprises, on the contrary, are not conducive to enterprise innovation and economic development. This proposition is in sharp contrast to the Porter hypothesis. Meanwhile, other scholars also showed that environmental protection policies would inhibit technological innovation. For example, both Zhao and Sun and Shi et al. claimed that environmental regulation would inhibit the efficiency of technological innovation [11,12]. Chen claimed that environmental regulation would inhibit the R&D investment (including the investment of research institutions, the investment of technology introduction and the investment of internal R&D) of industries with different pollution levels [13]. The third category is nonlinear effect. Some studies have showed that environmental regulation has a significant U-shaped relationship with enterprise innovation. With the strength of environmental regulation increasing, the enterprise innovation tends to decline first and then rise (e.g., [14,15]). Su and Zhou claimed that China’s formal environmental regulations would promote the innovation performance of firms with higher environmental standards, but inhibit the innovation performance of firms with lower environmental standards [16]. Long and Wan found that environmental regulation would promote the profit margin improvement of large-scale enterprises with low compliance costs and reduce the profit margin of small-scale enterprises with high compliance costs [17]. The fourth category is that there is no significant correlation between environmental policy and enterprise innovation. Kang et al. found that there was no significant correlation between incentive-based environmental regulation policies and enterprise innovation [18]. Sharif used the micro-enterprise data from 1983 to 1992 in the United States to study and found that there was no clear evidence to prove that environmental regulations would promote enterprise innovation [19].

According to the above analysis, much scholars used traditional indicators such as single pollutant discharge per unit output or pollution control expenditure to measure the intensity of environmental regulation. Different from previous studies, we use a new method to measure the environmental compliance index of enterprises by referring to the logic of accounting in measuring abnormal financial indicators. Specifically, the extent to which an enterprise complies with the government’s environmental laws and regulations is used to measure the intensity of environmental compliance. The larger the difference between the government’s environmental prescribed parameters and the enterprise’s environmental indicators, the smaller the enterprise’s environmental compliance index will be, and vice versa. The new measurement method of environmental compliance index constructed in this paper is an innovation in this research field.

In order to study the impact of environmental compliance on enterprise innovation, we build a mathematical model to theoretically analyze the impact of environmental compliance on enterprise innovation. According to the results of theoretical analysis, we propose not only hypothesis H_1_: Environmental compliance has a significant U-shaped relationship of first suppression and then promotion with enterprise innovation, but hypothesis H_2_: In terms of the overall effect, the promotion effect may play a leading role in the effect of environmental compliance on enterprise innovation in China’s industrial enterprises. In order to verify the hypothesis based on the theoretical analysis results, we use the random-effects Tobit model and the double hurdle model to empirically test the micro-data from China Industrial Enterprise Database from 1998 to 2013. Hence, by aiming at the improvement of the above points, we will have a certain degree of innovation. Firstly, based on the M-O theoretical model, we relax the original producer behavior hypothesis and embed environmental compliance factors and cost factors into the multi-product enterprise model, then, we construct a theoretical model applicable to environmental compliance and enterprise innovation. Secondly, based on previous studies, this paper refers to the logic of accounting to measure abnormal financial indicators and uses a new method to measure the environmental compliance index of enterprises. Finally, in terms of research methods, we use the random-effects Tobit model and the double hurdle model to empirically test the micro-data from China Industrial Enterprise Database from 1998 to 2013. Then, by using the Generalized Propensity Score Matching (GPSM) method to conduct robustness test. The robustness conclusion is drawn that the impact of environmental compliance on enterprise innovation shows a U-shaped relationship.

The structure of the remaining paper is organized as follows: Section 2 presents the theoretical model. Section 3 shows methodology and data, which illustrates the research methods and models, variable definition, data source and processing and descriptive statistical analysis. Section 4 describes empirical analysis and results, which illustrates benchmark results analysis, heterogenous texts and robustness test. Finally, Section 5 completes the conclusions and policy implications.

## 2. Theoretical Model

Different from previous studies on environmental regulations, this paper defines environmental compliance as the degree to which an enterprise’s production activities comply with environment-related regulations. Due to the different cognition and norms of different enterprises, much enterprises have different environmental compliance index. Based on M-O monopolistic competition and multi-product enterprise model, we relax the original producer hypothesis and embed the environmental compliance factor and cost factor into the multi-product enterprise equation. The equation is similar in theory to the equation proposed by Melitz and Ottaviano [20], but there are the entry of compliance factor and cost factor and the adjustment of entry cost, expected environmental compliance cost and expected environmental violation cost.

### 2.1. Consumer Equation

This paper retains the hypothesis of the equation proposed by Melitz and Ottaviano [20], that is, economic activity involves only one factor of production labor and two consumer goods. The first commodity is homogeneous, one unit of labor is needed to produce this commodity, and it is sold in perfect competition, and this commodity is chosen as money, so equilibrium wages are equal to 1. The other kind of commodity is horizontally differentiated (new product), continuously provided by *N* enterprises with returns on scale and monopolistic competition. Its preference is described by the following utility function:(1)$$U(q_{0};q(i),i\in [0,N])=\alpha \int_{0}^{N}q(i)di-\frac{\beta }{2}\int_{0}^{N}{\left[q(i)\right]}^{2}di-\frac{\gamma }{2}{\left[\int_{0}^{N}q(i)di\right]}^{2}+q_{0}$$
where q(i) is the quantity of variety i∈[0,N] and $q_{0}$ is the quantity of the homogeneous good. α is a measure of the intensity of preferences for the differentiated good with respect to the numéraire and may therefore be viewed as a measure of the size of the market, where γ expresses the substitutability between varieties: the higher γ, the closer substitutes the varieties. β>0 indicates that the representative consumers prefer the diversified consumption of their varieties.

According to the model hypothesis, the price equation is finally obtained:(2)p(i)=α−βq(i)−γQ
where $Q=\int_{0}^{N}q(j)dj$ represents the total market output of the differentiated product. *j* stands for different product categories.

### 2.2. Producter Equation

The cost involved in production equation mainly includes entry cost, fixed cost and marginal cost. Referring to the logic of Rocha et al. in measuring the labor compliance [21], this paper assumes that marginal cost can be divided into two parts. One part is that the marginal requirement of labor is assumed to be zero. The other part is that the marginal requirement of environmental compliance is assumed to be $\tau_{i}(h,t)$, and it consists of two parts. And the first part is the penalty cost faced by environmental non-compliance, and it is the increasing function of the possibility of being caught for environmental non-compliance. With the increase of h(•), the probability of being caught for environmental non-compliance and being punished will increase, and the increase of the possibility of being punished will lead to the expansion of environmental compliance costs. The second part is a function of the production time of an firm. In a short period of time, an firm must pay sunk costs if it wants to achieve certain environmental compliance. Hence, it is possible that the revenue brought by environmental compliance will be less than the cost of environmental compliance in the short term. This can explain the conclusion drawn by some scholars that the effect of environmental regulation on enterprise innovation is U-shaped. h(•) denotes the possibility of being caught by the government for environmental non-compliance, where (•) refers to the environmental compliance factor. The bigger h(•) is, the more likely the firm will be caught for environmental noncompliance. If h(•)=1 means that it is an established fact that the enterprise is caught, then the enterprise will lose its production capacity. If h(•)=0, it means that the firm cannot be caught and has full production capacity. We define 1−h(•) as environmental compliance index. The larger 1−h(•) is, the stronger environmental compliance will be, and the smaller expected marginal requirement will be faced by enterprises. We assume that the enterprise is faced with the same entry cost *G*, the expected marginal cost of production is $\tau_{i}(h,t)$, and the fixed cost of each type of product is *F*, then the total production cost of enterprise *i* is $G+\tau_{i}(h,t)+F$. To avoid losing the generality of the equation, we readjust the demand intercept α to zero, and the adjusted entry cost *G* becomes zero. The firm profit equation can be expressed as follows:(3)$$\Pi (i)=(1-h(•))p(i)q(i)-\tau_{i}(h,t)q(i)-F$$
where, $\tau_{i}(h,t)=C(1-h)+\phi /t(1-h)$ is a function of expected environmental compliance cost C(1−h), expected environmental violation cost ϕ/t(1−h) and time *t*, and ϕ is a constant. Expected environmental compliance cost C(1−h) is an increasing function of environmental compliance (1−h(•)). If time T is fixed, the expected marginal cost of a firm is negatively correlated with environmental compliance, that is, with the increase of environmental compliance, the expected cost of environmental violation ϕ/t(1−h) decreases.

Combining Equations (2) and (3), we can obtain the output $q^{*}$ and price $p^{*}$ under general equilibrium, and then the number of equilibrium producers $N^{*}$ is given by:(4)$$N^{*}=\frac{\sqrt{\beta /F(1-h(•))}\cdot [\alpha (1-h(•))-\tau_{i}(h,t)]-2\beta }{\gamma }$$

Entry and exit are free so that profits are zero in equilibrium. Thus, the optimal $q^{*}$ and $p^{*}$ are given by:(5)$$q^{*}=\frac{1}{\sqrt{\left(1-h(•)\right)}}\sqrt{F/\beta };p^{*}=\frac{1}{\sqrt{\left(1-h(•)\right)}}\sqrt{F\cdot \beta }$$

### 2.3. In Equilibrium

Combining $Q=Q_{i}+Q_{-i}=\int_{\omega \in \pi_{i}}q_{i}(\omega )d\omega +(M-1)\left|\pi \right|q$ and utility are maximized, where, ω represents the product category and *M* represents the number of enterprises with a given multi-product. $Q_{i}$ is the total output of firm i’s diversified products, and $Q_{-i}$ is the total output of firm i’s competitors. $\pi \subseteq R_{+}$, represents the collection of all product categories produced by enterprise *i* (*i* = 1......*M*), we get an equilibrium $p^{*}$ and $q^{*}$:(6)$$\left\{ \begin{array}{l} q^{*}=\frac{\alpha (1-h(•))-r(1-h)(M-1)\left|\pi \right|q-\tau_{i}(h,t)\left|\pi \right|}{2(1-h(•))(\beta +\gamma \left|\pi_{i}\right|)}\\ p^{*}=\frac{\alpha (1-h(•))-r(1-h)(M-1)\left|\pi \right|q+\tau_{i}(h,t)\left|\pi \right|}{2(1-h(•))} \end{array}\right.$$

This paper assumes that the cost for environmental compliance in the production process of the firm can be ignored, then the product and price functions of the firm are given by:(7)$$\left\{ \begin{array}{l} q^{*}=\frac{\alpha (1-h(•))-\gamma (1-h)(M-1)\left|\pi \right|q}{2(1-h(•))(\beta +\gamma \left|\pi_{i}\right|)}\\ p^{*}=\frac{\alpha (1-h(•))-\gamma (1-h)(M-1)\left|\pi \right|q}{2(1-h(•))} \end{array}\right.$$
at this point, we have the profit function:(8)Π(i)=(1−h(•))p(i)q(i)−F

Using (7) and (8), we have the product range of profit maximization, so that the product range of the M-th enterprise ${\left|\pi_{i}\right|}^{*}$ is given by Equation (9) (the specific solution process can be seen in the derivation of the Melitz and Ottaviano equation)
(9)$${\left|\pi_{i}\right|}^{*}=\frac{\alpha \sqrt{\left(1-h(•)\right)}\cdot \sqrt{\beta /F}-2\beta }{\gamma (M+1)}$$

According to Schumpeter’s innovation theory, the expansion of product range can measure the enhancement of innovation ability. Equation (9) shows that the product range ${\left|\pi_{i}\right|}^{*}$ produced by the enterprise will expand with the enhancement of environmental compliance 1−h(•), that is, with the enhancement of environmental compliance, the product range of the firm will expand accordingly. This is also consistent with the connotation of the Porter hypothesis.

In fact, ignoring the cost of environmental compliance is an absolute idealization that does not exist in real life. Therefore, considering $\tau_{i}(h,t)\neq 0$, the impact of environmental compliance on enterprise innovation depends on the size of expected environmental compliance cost C(1−h) and expected environmental violation cost ϕ/t(1−h). If C(1−h)≥ϕ/t(1−h), environmental compliance is not conducive to enterprise innovation, or even inhibits. If C(1−h)<ϕ/t(1−h), environmental compliance will promote enterprise innovation. Considering that enterprises need to buy a large amount of machinery and equipment in the early stage of environmental compliance, and the newly formulated punishment standards for environmental violation will not be too strict, the initial cost of environmental compliance of enterprises C(1−h) will be greater than the cost of environmental violation ϕ/t(1−h). In the later stage of environmental compliance, enterprises only need to improve their machinery and equipment, and the perfect environmental compliance standard will impose more and more strict penalties on enterprises violating environmental regulations. In this case, the cost of environmental compliance C(1−h) will be less than the cost of environmental violation ϕ/t(1−h). Therefore, we propose the first hypothesis:

**Hypothesis** **1** **(H1):**
*Environmental compliance has a significant U-shaped relationship of first suppression and then promotion with enterprise innovation.*


Given early implementation of environmental compliance regulations, most of China’s manufacturing companies may have already crossed the environmental compliance inflection point. In the 1980s, China began to explore the prevention and control of industrial pollution, including China’s first environmental protection conference held in 1982 and the subsequent “Trial Implementation of the Environmental Protection Law” and the “Several Decisions on Environmental Protection Work” promulgated in 1984. Therefore, we propose the second hypothesis:

**Hypothesis** **2** **(H2):**
*In terms of the overall effect, the promotion effect may play a leading role in the effect of environmental compliance on enterprise innovation in China’s industrial enterprises.*


## 3. Methodology and Data

### 3.1. Research Methods and Models

Considering the characteristics of the data set (in the Database of Chinese Industrial Enterprises, the dependent variable greater than 0 has a wide range of distribution, which can be regarded as a continuous variable whose condition depends on some factors and obeys normal distribution. In addition, the actual dependent variable cannot be negative, so deleting all the observed values of the dependent variable displayed as 0 will lead to biased estimation. Therefore, the Tobit model of random effects panel we chose is valid) and the fact that the Tobit model allows the estimated parameters to be unbiased and consistent, this paper uses the Tobit model to test the impact of environmental compliance on enterprise innovation. It is usually impossible for the nonlinear model of fixed effects to obtain a consistent estimate panel data [22]. Moreover, the preliminary test results of panel data of China’ industrial enterprises from 1998 to 2013 in this paper show that the maximum likelihood ratio rejects the original hypothesis that the mixed section Tobit model has no difference. In order to prevent spurious regression, we also conducted panel unit root test and co-integration test, and the results prove that the panel data used in this paper is stable and co-integration. Therefore, we select Tobit model of random effects panel. In view of the above analysis, used with Equation (9), we have the empirical model:(10)$$Pnov_{c,i,t}=\alpha +\beta_{1}H_{c,i,t-1}+\beta_{2}H_{c,i,t-1}^{2}+\gamma CV_{c,i,t-1}+\varepsilon_{c,i,t}$$
where, $Pnov_{c,i,t}$ represents the innovation level of enterprises in city *c*, industry *i* and year *t*, $H_{c,i,t-1}$ refers to 1 period lag environmental compliance. Considering the marginal requirement of environmental compliance $\tau_{i}(h,t)\neq 0$ and the non-linear relationship between environmental compliance and enterprise innovation confirmed by some scholars (e.g., [23]), we control the second term of 1 period lag environmental compliance $H_{c,i,t-1}^{2}$.$CV_{c,i,t-1}$ represents 1 period lag control variable, and $\varepsilon_{c,i,t}$ represents the random error term.

### 3.2. Variable Definition 

#### 3.2.1. Enterprise Innovation

According to existing studies, financial indicators related to new products, success rate of scientific research subjects or intellectual property data were chose as indicators of enterprise innovation performance. In this paper, enterprise innovation is measured for the proportion of sales revenue of new products in the sales revenue of enterprises, which according to the first-level index of the evaluation index system of enterprise independent innovation ability put forward by The National Bureau of Statistics of China. In order to ensure the robustness of the empirical model, we also use enterprises’ R&D investment as a substitute variable for enterprise innovation (*Pnov_c,i,_*_t_).

#### 3.2.2. Environmental Compliance Index

Environmental compliance refers to the degree to which an enterprise complies with the government’s environmental laws and regulations. The larger the difference between the government’s environmental regulation and the enterprise’s environmental indicators, the smaller the enterprise’s environmental compliance will be, and vice versa. This research assumes that the environmental expenses of a firm remain unchanged in the same region, industry and year. The mean environmental expenses of the region, industry and year are taken as the compliance standard, and the grouping scope is limited to the same region, industry and year. The main reason is that there are significant differences in the dimension of environmental costs of different firms. Although the assumption that “environmental costs of enterprises remain unchanged in the same region, industry and year” is strong, it conforms to the general equilibrium principle in reality [24]. Therefore, environmental compliance is measured by the difference between the proportion of sewage charge in product sales revenue of the enterprise and the average proportion of sewage charge in product sales revenue of the industry. According to the definition of environmental compliance (1−h(•)) in this research, if the difference between the actual average environmental cost of an enterprise and the actual average environmental cost of the industry is positive, it means that the probability of an enterprise being caught for non-compliance is zero, and then the compliance index 1−h(•) is 1. If the difference between the two is negative, it means that the probability of being caught for environmental noncompliance is greater than zero, and the smaller the negative number is, the more noncompliance the enterprise environment is. To this end, we use the logic of enterprise abnormal investment measurement () to standardize the data [25]. The specific formula of environmental compliance index after treatment can be written as follows:(11)$${\tilde{H}}_{k,i}=\left\{ \begin{array}{l} 1,H_{k,i}-{\bar{H}}_{i}\geq 0\\ 1+{\langle H_{k,i}-{\bar{H}}_{i}\rangle }^{standard},H_{k,i}-{\bar{H}}_{i}<0 \end{array}\right.$$
where ${\tilde{H}}_{k,i}$ represents the environmental compliance index, which is expressed as the difference between the proportion of pollutant discharge fee in product sales revenue of the firm and the proportion of pollutant discharge fee in product sales revenue of the industry (due to the lack of uniform parameters stipulated by the government, this research measures environmental compliance with the difference between the proportion of pollutant discharge fee in product sales revenue of an enterprise and the proportion of pollutant discharge fee in product sales revenue of an industry. Although this method has some endogenous problems, it can also reflect the environmental compliance intensity to a certain extent. For example, the proportion of pollutant discharge fee in product sales revenue of enterprise A is lower than the proportion of pollutant discharge fee in product sales revenue of the industry in which enterprise A is located. According to the assumption of this paper, enterprise A may be in violation of environmental compliance behaviors to A certain extent without considering reasonable avoidance of taxes. Moreover, the proportion of pollutant discharge fee in product sales revenue of enterprise A is lower than that of the industry in which enterprise A is located, and the higher the proportion of pollutant discharge fee in product sales revenue of the industry in which enterprise A is located, the greater the possibility that enterprise A has environmental compliance violations. In the empirical process, we used the robust standard error of aggregation at the industry level, and calculated the conventional robust standard error after controlling the industry fixed effect, that is, heteroscedasticity was allowed between groups, and the heteroscedasticity could be corrected to some extent by clustering within each group to reduce the variance). ${\bar{H}}_{i,j}$ represents the average value of the industry environmental compliance index, and *i* represents different industries. ${\langle H_{k,j,i}-{\bar{H}}_{i,j}\rangle }^{standard}$ represents the difference between the actual sewage charge of the enterprise after standardized treatment and the mean value of the industry (Table 1).

### 3.3. Data source and Processing

In view of the fact that the empirical research of this research needs to use indicators such as industrial enterprise value-added, product sales income, and industrial enterprise value-added, and in order to follow the completeness and continuity of the data, this paper selects the data of industrial enterprises above a certain scale the Database of China Industrial Enterprises from 1998 to 2013 (although the database of Chinese Industrial Enterprises has been updated to 2014 (the open data of The National Bureau of Statistics of China is only available until 2014), and there is no experts has yet used the 2014 data in view of the inconsistency between the 2014 data and the previous years’ data. In addition, the statistical data before 1998 were not complete, and there were many key variables missing about this research. Therefore, we selected the interval data from 1998 to 2013). We refer to intertemporal matching method of Brandt et al. and combine Generally Accepted Accounting Principles (GAAP) [27], the sample is deleted in one of the following situations: the enterprise code is missing; total assets, paid-up capital, net fixed assets, sales are missing or less than 0; the number of employees in the enterprise is less than 8; the total assets are less than current assets or less than the net value of fixed assets; the enterprise age is obviously wrong (the establishment year is behind the reporting year; the month is less than 1 or larger than 12); the enterprises are in abnormal business; the observed value of the main business income is less than 5 million yuan from 1998 to 2010 and the industrial enterprises with the main business income is less than 20 million yuan from 2011 to 2013. We eliminate the extreme value beyond the range of 5~95% of the key index. When there is no industrial output value (e.g., 2004), the calculation is carried out by using the industrial output value estimation formula [28]. We use interpolation method to complete the missing indicators from 1998 to 2013 (the concrete method of interpolation we used is as follows: if the enterprise appears in the previous year, the missing indicators will be supplemented according to the growth rate of product sales revenue. If the enterprise does not appear in the previous year, the missing indicators will be made up according to the growth rate of product sales revenue in the industry where the enterprise is located). Finally, the total number of effective samples obtained was 3,318,020, among which the sample size increased from 67,797 in 1998 to 259,468 in 2013.

In order to capture the unpredictable factors that change with the industry characteristics, we classify state-owned and non-state-owned manufacturing enterprises above designated size into 30 specific manufacturing industries according to the GB/T4754-2011 two-digit classification published by the National Bureau of Statistics of China, with 30 codes ranging from 13 to 42. We classify 30 specific manufacturing industries into three categories of labor-intensive, capital-intensive and technology-intensive, and the classification results are shown in Table 2 below. During the sample period, the corresponding industry code of industrial enterprises above the size has been adjusted for three times and finally adjusted to be the enterprise sample of the two-digit manufacturing industry in the industry classification standard of the national economy in 2011. Although the sample data only reaches 2013, the sample size is huge, and the panel data can reduce the endogeneity problem of empirical regression to some extent.

### 3.4. Descriptive Statistical Analysis

In order to capture the distribution of environmental compliance index in detail, we calculated the characteristics of the cleaned samples. The descriptive statistical results of the main variables are shown in Table 3. There were 3,317,975 observed samples of enterprise innovation, with a standard deviation of 0.083, which indicating that there was a small difference in the innovation level of the sample enterprises. The minimum value and maximum value are 0 and 0.339 respectively, which indicates that the innovation level of Chinese manufacturing enterprises is generally low, which also conforms to the practical inspiration of this goal we study to generally improve the innovation level of China’s enterprises. The standard deviation of environmental compliance in the first period after standardized treatment is 0.342, indicating that the compliance distribution of sample enterprises is relatively uniform. The minimum and maximum values of environmental compliance are 0 and 1, which indicating that the construction of environmental compliance indicators in this paper is in line with expectations. It is an established fact that enterprises with low environmental compliance (less than 0) will be caught. In this case, enterprises will lose production capacity, namely, h(•)→1. The standard deviation of the enterprise’s operating life and financing constraints is less than 1, and the distribution is relatively uniform. The descriptive statistics of the remaining variables are similar to previous studies, so it will not be repeated here.

## 4. Empirical Analysis and Findings

### 4.1. Benchmark Results Analysis

In order to test the rationality of random-effects Tobit model we constructed, we use the LR test to determine which estimation method is more suitable for the mixed Tobit model and the random-effects Tobit model. The test of LR rejects the null hypothesis (*p* value is 0), so there is an individual effect in enterprise innovation. The Tobit model of random effects panel adopted in this paper is more effective, and the basic regression results are shown in Table 4. We find that the empirical results of the innovation level decision-making of the two-column model in column (8) are basically the same as those in column (5), which confirms that the empirical model constructed in this paper is robust. According to the theory proposed by Scherer [29], we realize that compliance may have a nonlinear relationship with enterprise innovation, so columns (5) and (6) are regression results after controlling the quadratic term of the environment compliance ($H_{c,i,t-1}^{2}$). Therefore, column (5) has a more reasonable test result for controlling the second term of environment compliance. On the one hand, the regression coefficient of the quadratic term of compliance is relatively positive and significant at the 1% significance level. On the other hand, the 95% confidence interval of the inflection point calculated according to the columns (1) and (2) or the columns (5) and (6) is within the range of the independent variable. Therefore, the regression coefficient of the quadratic term of environmental compliance satisfies the assumption of the non-linear relationship between environmental compliance and enterprise innovation [30], and the impact of environmental compliance on enterprise innovation has a non-linear relationship. In order to further confirm the non-linear relationship of the impact of environmental compliance on corporate innovation, we use the three statistics of LM, LMF, and LRT to perform non-linear tests on columns (5) and (6), and all the results reject the null hypothesis that there is no non-linear relationship. According to the above analysis, environmental compliance has a significant non-linear relationship with corporate innovation, so there is no bias in the empirical model.

It is easy to find that the regression coefficient of 1 period lag operating life (*lnAge_c,i,t_*_-1_) is 0.001 and is significant at the 1% significance level with the observation in column (5). The reason seems that enterprises with long operating years are at a relatively mature stage of enterprise innovation, and their marginal cost of innovation is relatively low, so operating years are positively correlated with corporate innovation. The size of the firm (*lnSize_c,i,t_*_-1_) 1 period lag is significantly positively correlated with firm innovation, which indicates that the larger the firm, the higher its level of innovation. This is consistent with the conclusion of competition inhibition innovation under Schumpeter’s monopoly market research, that is, the larger the enterprise scale, the weaker the market competition effect, and the stronger the enterprise’s innovation ability. To our surprise, the regression coefficient of 1 period lag total factor productivity (*lnTfp_c,i,t_*_-1_) is significantly negative. The reason seems like factor market distortions lead to the inhibitory effect of total factor productivity on enterprise innovation, that is, market rent-seeking brought about by factor market distortions may cause enterprises with higher total factor productivity to invest in market rent-seeking activities rather than enterprise innovation activities. We suppose that total factor productivity has a nonlinear relationship with enterprise innovation. Therefore, we introduce the quadratic term of total factor productivity into the empirical model, and the empirical results confirm our conjecture that the influence of total factor productivity on enterprise innovation has a nonlinear relationship (where control 1 refers to the control of classifying state-owned and non-state-owned manufacturing enterprises above designated size into 30 specific manufacturing industries according to the two-digit classification published by the China’s National Bureau of Statistics. Control 2 refers to the control of labor-intensive, capital-intensive and technology-intensive three industry categories). The regression coefficient of 1 period lag current ratio (*Current_c,i,t_*_-1_) is significantly negative, and the possible reason is that excessive financing constraints lead to a greater debt pressure on enterprises and its lack of innovation motivation. The remaining control variables are in line with the expectations of this research, that is, the number of employees, per capita capital, export intensity, and subsidy income will all promote enterprise innovation.

The comparison results of column (5) and column (6) show that controlling the characteristics of different industries has little effect on the overall regression results (where control 1 refers to the control of classifying state-owned and non-state-owned manufacturing enterprises above designated size into 30 specific manufacturing industries according to the two-digit classification published by the China’s National Bureau of Statistics. Control 2 refers to the control of labor-intensive, capital-intensive and technology-intensive three industry categories), and the regression model is relatively robust. The empirical results of column (5) in Table 4 indicate that the regression coefficient of the first-stage environmental compliance item is significantly negative, and the regression coefficient of the second-order environmental compliance item is significantly positive. The inflection point calculated according to column (5) is 0.669, and its 95% confidence interval is within the range of the independent variable. This indicates that environmental compliance has a U-shaped relationship with enterprise innovation. The environmental compliance will inhibit corporate innovation, when the enterprise environmental compliance index is lower than 0.669.

Taking the empirical results of column (5) in Table 4 as an example, we conducted a more in-depth analysis of the distribution characteristics of specific enterprises and industries around the inflection point of environmental compliance. Firstly, the sample of companies on the left of the inflection point of environmental compliance is 925,268, accounting for 26.53% of the total number of companies, and the sample of companies on the right of the inflection point of environmental compliance is 2,562,702, accounting for 73.47% of the total number of companies. Therefore, in terms of the effect of environmental compliance of China’s manufacturing enterprises above designated size on enterprise innovation, environmental compliance has a promoting effect on enterprise innovation in a relatively large number of enterprise samples, and it has an inhibitory effect on corporate innovation in a relatively small sample of industries. To some extent, this can be interpreted as, from the overall effect, the promotion effect may play a leading role in the impact effect of environmental compliance on enterprise innovation of industrial enterprises above a large scale in China. Secondly, the proportion of enterprises on both sides of the inflection point of environmental compliance in the total number of enterprises in the industry in Figure A1 (Appendix A) shows that the sample industries in which environmental compliance has a promoting effect on enterprise innovation are mainly distributed in labor-intensive and capital-intensive industries. However, there is a polarization in technology-intensive industries. For instance, the distribution of manufacturing of chemical raw materials and chemical products, pharmaceutical manufacturing and chemical fiber manufacturing and other technology-intensive industries is relatively large. In contrast, the sample industries in which environmental compliance has a inhibition effect on enterprise innovation are mainly distributed in automobile manufacturing, electrical machinery and equipment manufacturing, and computer, communications and other electronic equipment manufacturing and other technology-intensive industries, and relatively less in labor-intensive and capital-intensive industries. The comparison results further indicate that from the point of view of China’s current development facts, China’s industrial sector is in the process of transformation and upgrading as the technology-intensive manufacturing industry develops and grows. Although governments at all levels will give various guidance and incentive policies to technology-intensive manufacturing enterprises to some extent, half of technology-intensive industries, including automobile manufacturing, electrical machinery and equipment manufacturing, and computer, communications and other electronic equipment manufacturing, still face the low environmental compliance index trap.

### 4.2. Heterogenous Texts

In order to investigate the impact of environmental compliance on enterprise innovation in heterogeneous enterprises, this research divides enterprise samples into state-owned enterprises and non-state-owned enterprises according to the structure of enterprise property rights. According to the enterprise’s operating years, whether the enterprise is located in the core area and whether the enterprise exports, the enterprise samples are divided into mature enterprises and non-mature enterprises, core enterprises and non-core enterprises, and export enterprises and non-export enterprises. The results of heterogeneity test about environmental compliance affecting enterprise innovation are shown in Table 5.

The primary coefficients of environmental compliance of different types of enterprises are all significantly negative, and the secondary coefficients of environmental compliance are all significantly positive. Specifically, the primary environmental compliance coefficients of state-owned enterprises, mature enterprises, core area enterprises, and export enterprises are all lower than the primary environmental compliance coefficients of non-state-owned enterprises, immature enterprises, non-core area enterprises, and non-export enterprises. This means that among companies with lower levels of environmental compliance, environmental compliance has a greater inhibitory effect on the innovation of state-owned enterprises, mature enterprises, core area enterprises and export enterprises. The possible reason is that compared with non-state-owned enterprises, immature enterprises, non-core area enterprises and non-export enterprises, state-owned enterprises, mature enterprises, core area enterprises and export enterprises tend to have larger scale and more sectors. Therefore, these types of enterprises below the inflection point of environmental compliance often need to spend huge costs to purchase large-scale machinery and equipment to reduce environmental pollution. Therefore, the environmental compliance cost C(1−h) is relatively high, and environmental compliance has a greater inhibitory effect on these types of enterprise innovation.

Compared with the non-state-owned enterprises, immature, enterprises in non-core areas and non-export enterprises, state-owned enterprises, mature, enterprises in core areas (according to China’s “Seventh Five-year Plan” adopted at the fourth Session of the sixth National People’s Congress, the core areas refer to the 11 provinces (cities) in the eastern region, including Beijing, Tianjin, Hebei, Liaoning, Shanghai, Jiangsu, Zhejiang, Fujian, Shandong, Guangdong and Hainan, while the non-core areas refer to the provinces (cities) in the central and western regions) and export enterprises of environmental compliance quadratic term coefficient is larger, which means environmental compliance had a greater promoting effect on innovation about state-owned enterprises, mature enterprises, enterprises in core areas and export enterprises over the inflection point of enterprise environmental compliance. The possible reasons are as follows: First, compared with state-owned enterprises, it is difficult for non-state-owned enterprises to overcome the bottleneck of enterprise innovation financing, which may lead to the lack of available innovation capital for state-owned enterprises, and the low innovation efficiency leads to the small promoting effect of environmental compliance on enterprise innovation [31,32]. Therefore, beyond the inflection point of environmental compliance, environmental compliance of state-owned enterprises has a great promotion effect on enterprise innovation. Second, compared with immature enterprises, mature enterprises are already in the relatively mature stage of enterprise innovation, and their innovation marginal cost is relatively low. Therefore, under the condition of constant marginal rate of return, the optimal innovation investment increases, and more funds are invested in existing innovation projects, so that the contribution of environmental compliance investment to the innovation of mature enterprises is greater than its contribution to non-mature enterprises. Third, compared with enterprises in non-core areas, enterprises in core areas are faced with higher marketization level, stricter market access and negative list, and their environmental violation costs will be higher than those of enterprises in non-core areas. Therefore, beyond the inflection point of environmental compliance, the promoting effect of environmental compliance on enterprise innovation in core areas is greater than that of enterprises in non-core areas. Fourth, on the one hand, according to the export self-selection mechanism of Melitz [33], the enterprises that take the lead in export have higher productivity and yield rate. The export enterprises obtain the research and development (R&D) funds from operating profit are greater than those non-export enterprises, and it is easier for export enterprises to obtain various innovation resources from the outside. On the other hand, export enterprises face stricter and more severe international environmental compliance rules, and the cost of enterprise violation is far greater than the cost of enterprise environmental compliance construction. Therefore, the environmental compliance of export enterprises with high environmental compliance contributes more to enterprise innovation.

### 4.3. Robustness Test

To further prove the robustness of the empirical model, we incorporated 1 period lag enterprise innovation (*Pnov_c,i,t_*_-1_) into the random-effects Tobit model, and used 1 period lag ratio of enterprises’ R&D investment to product sales revenue (*RD_c,i,t_*_-1_) as an alternative variable of enterprise innovation. To reflect the causal impact of environmental compliance on enterprise innovation better and eliminate measurement errors caused by heterogeneity between the treatment group and the control group before receiving policy treatment, this research uses a Generalized Propensity Score Matching method (GPSM) developed by Hirano and Imbens [34], which is based on continuous treatment of variables, to further test the impact of different environmental compliance levels on enterprise innovation.

The empirical results of 1 period lag enterprise innovation (*Pnov_c,i,t_*_-1_) are shown in column (3) in Table 6. The empirical results of introducing 1 period lag enterprise innovation into column (3) are consistent with the random-effects Tobit model of column (1) and the double hurdle model of column (2) after controlling the control variables and industry characteristics. We use the 1 period lag ratio of R&D investment to product sales revenue (*RD_c,i,t_*_-1_) as an alternative variable of enterprise innovation in column (4), and the empirical results are basically consistent with the column (1) and column (2), and it indicates that the empirical results of this research are relatively robustness.

## 5. Conclusions and Policy Implications

### 5.1. Conclusions

This paper embeds environmental compliance factors and compliance cost factors into the M-O monopolistic competition and multi-product enterprise model to construct a theoretical model applicable to environmental compliance and enterprise innovation. In addition, we also construct a new environmental compliance index. We use the random-effect Tobit model and double hurdle model to empirically test the micro-data of China’s manufacturing enterprises from 1998 to 2013, then we use the GPSM to conduct a robustness test. The robustness conclusion is that environmental compliance has a significant U-shaped relationship with enterprise innovation, that is, environmental compliance will inhibit enterprise innovation on the left of the inflection point of environmental compliance (0.669), while environmental compliance on the right of the inflection point will promote enterprise innovation. This is an important finding that is different from the existing researches, which reveals the complex and nonlinear impact of environmental compliance on enterprise innovation. The unique meaning of this nonlinear effect is that only when the environmental compliance index reaches a certain level can it have a promoting effect on enterprise innovation. Otherwise, the environmental compliance behavior of enterprises will have a significant crowding out effect on enterprise innovation. We also conduct empirical tests based on further consideration of enterprise heterogeneity factors. It is found that environmental compliance has greater inhibitory effects on innovation of state-owned enterprises, mature enterprises, core areas enterprises and export enterprises among enterprises with a lower level of environmental compliance, but environmental compliance has a greater promoting effect on state-owned enterprises, mature enterprises, core areas enterprises and export enterprises, when it beyond the inflection point of enterprise environmental compliance.

### 5.2. Policy Implications

The research conclusion provides the following policy implications. Firstly, the environmental compliance measurement system should be established. Due to the environmental compliance is a legal issue regulated through “yes” or “no” condition, the government should focus its supervision on “no” enterprises, i.e., those with an environmental compliance less than 1. Therefore, based on the environmental compliance measurement index constructed in this research, it is necessary to further improve the measurement system of environmental compliance and incorporate more environmental pollution indicators into the measurement system. The government focuses on regulating various types of companies whose indicators exceed those permitted by the license of that particular firm. On the other hand, firms need to make adjustments according to various indicators of the environmental compliance in order to fulfill their obligations of environmental laws and regulations. Secondly, the government should implement environmental compliance construction step by step. The government should adopt progressive policies to promote environmental compliance construction, and search the inflection point of the impact of current environmental compliance on corporate innovation, so as to stimulate the role of environmental compliance in promoting micro-enterprise innovation. And the government should subsidy the technology-intensive enterprises at low level of environmental compliance and support the innovation of technology-intensive enterprises in breakthrough of key core technologies. Thirdly, the government should strengthen environmental compliance supervision of labor-intensive and capital-intensive enterprises. Because of increasing the punishment for violation can further improve the environmental compliance index of labor-intensive and capital-intensive enterprises by increasing the cost of environmental violation, hence, expanding the transformation and upgrading of labor-intensive and capital-intensive enterprises. Finally, the government should encourage heterogeneous enterprises to enhance compliance construction. The environmental compliance has greater promoting effects on state-owned enterprises, mature enterprises, core enterprises and export enterprises beyond the inflection point of enterprise environmental compliance. Therefore, when innovative resources are completely allocated by the market, there will inevitably be market failure in non-state-owned enterprises, non-mature enterprises, enterprises in non-core areas and non-export enterprises without innovation resources. Therefore, the government should give certain innovative preferential policies to non-state-owned enterprises, non-mature enterprises, enterprises in non-core areas and non-export enterprises.

## Figures and Tables

**Table 1 ijerph-18-01924-t001:** Usage description of main variables.

Variable Types	Variable Code	Variable Name	Explanation	Expected Symbol
Explained variable	Enterprise innovation	*Pnov_c,i,t_*	The ratio of new product sales revenue to the total product sales revenue.	
Explanatory variables	Environmental compliance index	*H_c,i,t-_* _1_	Calculated using the calculation equation in this research. See Equation (11) for details, and 1 period lag.	+
Control variables	Enterprise size	*lnSize_c,i,t-_* _1_	The total fixed assets take logarithm, and 1 period lag.	+
	Total factor productivity (TFP)	*lnTfp_c,i,t-_* _1_	LP method is used to calculate, and 1 period lag.	+
	The age of operation	*lnAge_c,i,t-_* _1_	The age of enterprise operation take logarithm, and 1 period lag.	+
	The number of employees	*lnL_c,i,t-_* _1_	The number of employees in an enterprise takes the logarithm, and 1 period lag.	+
	Asset-labor ratio	*lnKL_c,i,t_* _-1_	The ratio of assets to labor is logarithmic, and 1 period lag.	+
	Current ratio	*Current_c,i,t_* _-1_	The ratio of current assets to current liabilities, and 1 period lag.	+/−
	Financing constraints	*Fin_c,i,t_* _-1_	The ratio of interest expense to fixed asset net value, and 1 period lag.	+
	Subsidies income	*lnSubsidy_c,i,t_* _-1_	The ratio of Subsidy income to sales income ratio, and 1 period lag.	+
	Export intensity	*lnExport_c,i,t_* _-1_	Take the logarithm of export delivery value, and 1 period lag.	+

Source: The Database of China Industrial Enterprises (With the total factor productivity estimation method put forward by Berry, Levinsohn and Pakes (LP method for short) [26], LP method uses the intermediate input variable of the firm as the adjustable factor input when the firm is impacted by productivity. The total factor productivity measured by the LP method can be expressed as: $tfp_{it}^{LP}=Y-{\hat{\beta }}_{it}^{LP}-{\hat{\beta }}_{l}^{LP}L_{it}-{\hat{\beta }}_{k}^{LP}K_{it}-{\hat{\beta }}_{m}^{LP}M_{it}-\eta_{it}$. We use industrial added value to replace the output of enterprises, and use the producer price index of products in different industries for the reduction. Referring to the treatment method of Brandt et al., we use the perpetual inventory method to calculate the capital stock in order to achieve detailed and accurate TFP estimation).

**Table 2 ijerph-18-01924-t002:** Industry classification by factor intensity (30 industries).

Labor-Intensive Industry	Capital-Intensive Industry	Technology-Intensive Industry
Agricultural and sideline food processing industry (13) Food manufacturing industry (14) Wine, beverage and refined tea manufacturing industry (15) Tobacco Products industry (16) Textile industry (17) Textile clothing, clothing industry (18) Leather, fur, feather and their products and footwear (19) Wood processing and wood, bamboo, rattan, palm, grass products industry (20) Furniture manufacturing (21) Paper and paper products industry (22) Printing and recording media reproduction industry (23) Culture and education, industrial beauty, sports and entertainment manufacturing (24) Rubber and plastic products (29) Other manufacturing industries (41) Comprehensive utilization of waste resources (42)	Petroleum processing, coking and nuclear fuel processing industries (25) Non-metallic mineral products industry (30) Ferrous metal smelting and calendering industry (31) Non-ferrous metal smelting and calendering industry (32) Metal products industry (33) General equipment manufacturing (34) Special equipment manufacturing (35) Manufacturing of railway, shipping, aerospace and other transportation equipment(37) Instrumentation and culture, office machinery manufacturing (40)	Manufacturing of chemical raw materials and chemical products (26) Pharmaceutical manufacturing (27) Chemical fiber manufacturing (28) Automobile manufacturing industry (36) Electrical machinery and equipment manufacturing (38) Manufacturing of computers, communications and other electronic equipment (39)

Source: China Industry Business Performance Data.

**Table 3 ijerph-18-01924-t003:** Descriptive statistics of main variables.

Variables	Observations	Mean	Standard Deviation	Minimum	Maximum
*Pnov_c,i,t_*	3,317,975	0.027	0.083	0	0.339
*H_c,i,t_* _-1_	2,586,771	0.736	0.342	0	1
*lnAge_c,i,t_* _-1_	2,506,420	1.869	0.880	0	7.602
*lnSize_c,i,t_* _-1_	2,574,927	8.695	1.708	−0.095	18.94
*lnTfp_c,i,t_* _-1_	2,563,855	8.154	1.033	2.957	16.54
*lnL_c,i,t_* _-1_	2,585,580	4.930	1.108	0	12.29
*lnKL_c,i,t_* _-1_	2,563,874	5.164	1.678	−9.879	9.245
*Current_c,i,t_* _-1_	2,537,542	2.157	5.624	0.001	64.09
*Fin_c,i,t_* _-1_	2,486,844	0.066	0.167	−0.056	1.633
*lnSubsidy_c,i,t_* _-1_	2,255,394	0.787	2.070	0	15.39
*lnExport_c,i,t_* _-1_	2,274,359	2.974	4.575	0	19.04

Note: Except the dependent variable, all the other variables 1 period lag.

**Table 4 ijerph-18-01924-t004:** Environmental compliance and enterprise innovation: Benchmark results.

Variable	Random-Effect Tobit Model						Double Hurdle	
	(1)	(2)	(3)	(4)	(5)	(6)	Willingness	Level
*H_c,i,t_* _-1_	−0.114 ***	−0.119 ***	−0.103 ***	−0.104 ***	−0.122 ***	−0.128 ***	−1.448 ***	−0.194 ***
	(0.001)	(0.001)	(0.000)	(0.000)	(0.001)	(0.001)	(0.025)	(0.008)
*H_c,i,t_* _-1_ ^2^	0.109 ***	0.113 ***			0.091 ***	0.096 ***	1.251 ***	0.145 ***
	(0.001)	(0.001)			(0.001)	(0.001)	(0.021)	(0.006)
*lnAge_c,i,t_* _-1_			0.001 ***	0.001 ***	0.001 ***	0.001 ***	0.144 ***	−0.001
			(≈0)	(≈0)	(≈0)	(≈0)	(0.002)	(0.001)
*lnSize_c,i,t_* _-1_			0.002 ***	0.002 ***	0.002 ***	0.002 ***	0.043 ***	−0.006 ***
			(≈0)	(≈0)	(≈0)	(≈0)	(0.001)	(0.000)
*lnTfp_c,i,t_* _-1_			−0.004 ***	−0.004 ***	−0.004 ***	−0.004 ***	−0.088 ***	−0.002 ***
			(≈0)	(≈0)	(≈0)	(≈0)	(0.0017)	(0.001)
*lnL_c,i,t_* _-1_			0.002 ***	0.002 ***	0.002 ***	0.002 ***	0.176 ***	0.001 *
			(≈0)	(≈0)	(≈0)	(≈0)	(0.002)	(0.001)
*lnKL_c,i,t_* _-1_			0.011 ***	0.011 ***	0.011 ***	0.011 ***	0.100 ***	0.012 ***
			(≈0)	(≈0)	(≈0)	(≈0)	(0.001)	(0.000)
*Current_c,i,t_* _-1_			0.000 ***	0.000 ***	0.000 ***	0.000 ***	−0.005 ***	−0.000 **
			(≈0)	(≈0)	(≈0)	(≈0)	(0.000)	(≈0)
*Fin_c,i,t_* _-1_			−0.033 ***	−0.034 ***	−0.034 ***	−0.034 ***	0.239* **	−0.03 8***
			(0.000)	(0.000)	(0.000)	(0.000)	(0.009)	(0.003)
*lnSubsidy_c,i,t_* _-1_			0.001 ***	0.001 ***	0.001 ***	0.001 ***	0.041 ***	0.002 ***
			(≈0)	(≈0)	(≈0)	(≈0)	(0.001)	(0.000)
*lnExport_c,i,t_* _-1_			0.001 ***	0.001 ***	0.001 ***	0.001 ***	0.022 ***	0.000
			(≈0)	(≈0)	(≈0)	(≈0)	(0.000)	(≈0)
Constant	0.011 ***	0.020 ***	0.013 ***	0.021 ***	0.016 ***	0.024 ***	−2.459 ***	0.224 ***
	(0.000)	(0.000)	(0.001)	(0.001)	(0.001)	(0.001)	(0.0130)	(0.004)
Industry characteristics	Control 1	Control 2	Control 1	Control 2	Control 1	Control 2	Control 1	Control 2
/sigma_u	0.050 ***	0.050 ***	0.055 ***	0.055 ***	0.055 ***	0.055 ***		
	(≈0)	(≈0)	(≈0)	(≈0)	(≈0)	(≈0)		
/sigma_e	0.047 ***	0.047 ***	0.049 ***	0.049 ***	0.049 ***	0.049 ***		
	(≈0)	(≈0)	(≈0)	(≈0)	(≈0)	(≈0)		
sigma								0.151 ***
								(0.0003)
*ρ*	0.534	0.532	0.558	0.556	0.558	0.555		
Log-L	3,707,020.7	3,623,543.7	2,785,268.5	2,721,770.2	2,785,418.5	2,721,995.2	3,163,286.3	3,164,526.4
Observations	2,586,729	2,531,584	2,023,259	1,979,583	2,023,259	1,979,583	2,418,112	2,418,112

Note: Both explanatory variables and control variables 1 period lag, where *H_c,i,t_*_-1_^2^ represent the second-order 1 period lag environmental compliance items. This research adopts the robustness standard errors aggregated at the industry level. ***, **, * indicate significant at 1%, 5%, and 10% respectively, and the robust standard error is reported in parentheses, the same below.

**Table 5 ijerph-18-01924-t005:** Environmental compliance and enterprise innovation: Heterogeneity test results.

Variables	(1)	(2)	(3)	(4)	(5)	(6)	(7)	(8)
	State-Owned	Non-State-Owned	Mature	Immature	Core Areas	Non-Core Areas	Export	Non-Export
*H_c,i,t_* _-1_	−0.123 ***	−0.114 ***	−0.123 ***	−0.114 ***	−0.122 ***	−0.121 ***	−0.133 ***	−0.118 ***
	(0.001)	(0.002)	(0.001)	(0.002)	(0.001)	(0.003)	(0.002)	(0.001)
*H_c,i,t_* _-1_ ^2^	0.117 ***	0.112 ***	0.117 ***	0.112 ***	0.117 ***	0.116 ***	0.123 ***	0.116 ***
	(0.001)	(0.002)	(0.001)	(0.002)	(0.001)	(0.002)	(0.002)	(0.001)
Constant	0.014 ***	−0.005 ***	0.014 ***	−0.005 ***	0.020 ***	0.002	0.002	0.019 ***
	(0.001)	(0.001)	(0.001)	(0.001)	(0.001)	(0.002)	(0.002)	(0.001)
Control variables	Control	Control	Control	Control	Control	Control	Control	Control
Industry characteristics	Control 1	Control 1	Control 1	Control 1	Control 1	Control 1	Control 1	Control 1
sigma_u	0.055 ***	0.054 ***	0.055 ***	0.054 ***	0.054 ***	0.058 ***	0.064 ***	0.055 ***
	(≈0)	(0.0001)	(≈0)	(0.000)	(≈0)	(0.000)	(0.000)	(≈0)
sigma_e	0.050 ***	0.041 ***	0.050 ***	0.041 ***	0.048 ***	0.051 ***	0.059 ***	0.041 ***
	(≈0)	(≈0)	(≈0)	(≈0)	(≈0)	(≈0)	(≈0)	(≈0)
ρ	0.552	0.632	0.552	0.632	0.552	0.567	0.536	0.641
Log-L	2,082,950.3	684,766.71	2,082,950.3	684,766.71	2,173,377.8	615,479.08	744,259.07	1,963,919.6
Observations	1,535,020	488,239	1,535,020	488,239	1,560,682	462,577	621,838	1,323,456

*** indicate significant at 1%.

**Table 6 ijerph-18-01924-t006:** Environmental compliance and enterprise innovation: Robustness.

Variables	(1)	(2)	(3)	(4)	(5)
	Random-Effect Tobit Model	Double Hurdle Model	Random-Effect Tobit Model	Random-Effect Tobit Model	GPSM
*Pnov_c,i,t_* _-1_			0.564 ***		
			(0.001)		
*H_c,i,t_* _-1_	−0.122 ***	−0.194 ***	−0.140 ***	−0.995 ***	
	(0.001)	(0.008)	(0.001)	(0.022)	
*H_c,i,t_* _-1_ ^2^	0.091***	0.145 ***	0.10 5***	0.151 ***	
	(0.001)	(0.006)	(0.001)	(0.019)	
*Pscore_c,i,t_* _-1_					−0.121 ***
					(0.001)
*Pscore_c,i,t_* _-1_ ^2^					0.118 ***
					(0.001)
Constant	0.224 ***	0.224 ***	−0.016 ***	0.361 ***	−0.005 ***
	(0.004)	(0.004)	(0.000)	(0.016)	(0.001)
Control variables	Control	Control	Control	Control	Control
Industry characteristics	Control 1	Control 1	Control 1	Control 1	Control 1
/sigma_u	0.055 ***		0.0137 ***	3.075 ***	0.056 ***
	(≈0)		(0.000)	(0.003)	(≈0)
/sigma_e	0.049 ***		0.052 ***	0.791 ***	0.049 ***
	(≈0)		(≈0)	(0.000)	(≈0)
sigma		0.151***			
		(0.0003)			
*ρ*	0.558		0.065	0.940	0.492
Log-L	2,785,418.5	3,164,526.4	−3,511,470.2	−3,443,592	1,894,004.5
Observations	2,023,259	2,418,112	2,023,259	1,991,203	1,389,267

Note: Where *Pscore_c,i,t_*_-1_ is the environmental compliance agent variable matched by GPSM method, and *Pscore_c,i,t_*_-1_^2^ is the quadratic terms of environmental compliance matched by GPSM method. *** indicate significant at 1%.

## Data Availability

The datasets used and/or analysis during the current study are available from the corresponding author on reasonable request.

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
