# Peer review of "Environmental Compliance and Enterprise Innovation: Empirical Evidence from Chinese Manufacturing Enterprises"

_ijerph, 2021, doi:10.3390/ijerph18041924_

Round 1

Reviewer 1 Report

  • The description of the methods would be difficult to follow for the targeted practitioner community, i.e. the people who would actually use the method and results. The authors are, therefore, encouraged to add a very simple illustration of the method and its application before going on to the main case study involving real enterprises. In fact, the paper could even be split into two papers: one introducing the theoretical methodology with a very simple case study, and another paper conducting a thorough case study on real enterprises.
  • The explanation of the environmental compliance index needs better and clearer explanation in words.
  • The main practical conclusion seem to be that due to the U shape of relationship between environmental compliance and innovation, the Government should adopt different policies for different enterprises depending on which side of the U they fall. This seems reasonable.
  • The difficulty, however, is environmental compliance is a legal matter regulated through Yes and No conditions. For example, are the emissions of a specific pollutant above or below what is allowed by the permit for the particular company? How could an enterprise wishing to use the method from this work combine it with their legal obligations under environmental laws? The authors should discuss this carefully.
  • The dark and light grey colors on Figure 1a appear to be inverted, i.e. the text at the top does not seem to be consistent with the figure.

Reviewer 2 Report

Overview:

The paper examines the relationship between environmental compliance and enterprise innovation in China’s manufacturing sector. The authors establish a theoretical framework to analyse this relationship by employing the M-O monopolistic competition and multi-product enterprise model. Based on this theoretical underpin, an empirical analysis is carried out by using the random-effects Tobit model and the double hurdle model with the enterprise-level micro panel data from 1998 to 2013. The paper is well written, and the techniques are appropriate and adequate for the analysis of this issue. I see no obvious weaknesses in the methodology of the analysis. Therefore, my general view is that the paper could be considered for publication with some minor revisions. I would like to draw the attention of the authors to the following specific comments which could be considered in revising or strengthening the paper.

Specific comments:

  1. Section 1 Introduction could be improved by further explanation of the differences in the U-shaped relationship between environmental compliance and enterprise innovation claimed in the third category of existing literature (e.g. Yang and Liu, 2020 etc.) and this paper (Ln 99-101). Is the third category of studies the most obvious antecedent of your work?

  1. Ln 113-14, Section 2, the authors state: ‘… there are some important entries and the innovation of production structure.’ This needs to be elaborated further for readers to understand better the meaning of this sentence, especially ‘some important entries’ and ‘the innovation of production structure’.

  1. Ln 211-13, Section 3.1, it is not clear if the unit root test and the cointegration test have been carried out before the Tobit model is used in empirical estimations.

  1. For the calculation of environmental compliance index by using Equation 11 (Ln 247-54), the measured environmental compliance index is expressed as the difference between the proportion of pollutant discharge fee in product sales revenue of the enterprise and the proportion of pollutant discharge fee in product sales revenue of the industry. One could argue that it would be better to employ the standard issued by the state (if it is available) instead of using the average value of the industry here. In Footnote 3, reader would be interested to know how and to what extent the endogeneity problem can be reduced by adopting the standard error of robustness aggregated at the industry level.

  1. The paper would benefit from a thorough proofreading. Basic grammar is for the most part adequate, except for some terms or jargons used in the paper which are not English language usage or idiom and not easy to understand. This defect can be cured only if the manuscript is submitted to a fully competent expert in the English language for a thorough proofreading.

I thought I could simply list some required corrections, but there are some others I barely skimmed the surface, as per following examples.

Ln 51, 59, 70, ‘The first type’, ‘The second type’ and ‘The third type’ should be changed to ‘category’ which is consistent with ‘The fourth category’ in Ln 81

Ln 102, ‘…the remainder of this document…’ should be ‘the remainder of this paper…’; ‘The second part’ should be ‘Section 2’ etc.

Readers would be struggling to comprehend terms and expressions below used in the paper. 

Ln 68-9, ‘… expenditures expenditure of research institutions…’

Ln 75-6, ‘… the ministry of Industry of enterprises…’

Ln 87-8, ‘…the impact of environmental compliance on enterprise innovation exists in various situations.’

Ln 113-14, ‘… there are some important entries and the innovation of production structure.’

Ln 130-31, ‘… hypothesis and theoretical,…’

Ln 202, ‘Empirical preparation’, should it be ‘Methodology and Data’?

Ln 273, ‘…missing research indicators…’

… and so on

Reviewer 3 Report

Comment are presented in the attached file.
